# Characterization of the O-Glycoproteome of *Porphyromonas gingivalis*

Paul D. Veith,[a] ⓘ Mikio Shoji,[b] ⓘ Nichollas E. Scott,[c] ⓘ Eric C. Reynolds[a]

[a]Oral Health Cooperative Research Centre, Melbourne Dental School, Bio21 Institute, University of Melbourne, Parkville, Victoria, Australia
[b]Department of Microbiology and Oral Infection, Graduate School of Biomedical Sciences, Nagasaki University, Nagasaki, Japan
[c]Department of Microbiology and Immunology, University of Melbourne at the Peter Doherty Institute for Infection and Immunity, Melbourne, Australia

**ABSTRACT** *Porphyromonas gingivalis* is an important human pathogen and also a model organism for the Bacteroidetes phylum. O-glycosylation has been reported in this phylum with findings that include the O-glycosylation motif, the structure of the O-glycans in a few species, and an extensive O-glycoproteome analysis in *Tannerella forsythia*. However, O-glycosylation has not yet been confirmed in *P. gingivalis*. We therefore used glycoproteomics approaches including partial deglycosylation with trifluoromethanesulfonic acid as well as both HILIC and FAIMS based glycopeptide enrichment strategies leading to the identification of 257 putative glycosylation sites in 145 glycoproteins. The sequence of the major O-glycan was elucidated to be HexNAc-HexNAc(P-Gro-[Ac]$_{0-2}$)-dHex-Hex-HexA-Hex(dHex). Western blot analyses of mutants lacking the glycosyltransferases PGN_1134 and PGN_1135 demonstrated their involvement in the biosynthesis of the glycan while mass spectrometry analysis of the truncated O-glycans suggested that PGN_1134 and PGN_1135 transfer the two HexNAc sugars. Interestingly, a strong bias against the O-glycosylation of abundant proteins exposed to the cell surface such as abundant T9SS cargo proteins, surface lipoproteins, and outer membrane $\beta$-barrel proteins was observed. In contrast, the great majority of proteins associated with the inner membrane or periplasm were glycosylated irrespective of their abundance. The *P. gingivalis* O-glycosylation system may therefore function to establish the desired physicochemical properties of the periplasm.

**IMPORTANCE** *Porphyromonas gingivalis* is an oral pathogen primarily associated with severe periodontal disease and further associated with rheumatoid arthritis, dementia, cardiovascular disease, and certain cancers. Protein glycosylation can be important for a variety of reasons including protein function, solubility, protease resistance, and thermodynamic stability. This study has for the first time demonstrated the presence of O-linked glycosylation in this organism by determining the basic structure of the O-glycans and identifying 257 glycosylation sites in 145 proteins. It was found that most proteins exposed to the periplasm were O-glycosylated; however, the abundant surface exposed proteins were not. The O-glycans consisted of seven monosaccharides and a glycerol phosphate with 0–2 acetyl groups. These glycans are likely to have a stabilizing role to the proteins that bear them and must be taken into account when the proteins are produced in heterologous organisms.

**KEYWORDS** *Porphyromonas gingivalis*, O-glycosylation, glycoproteins, proteome, trifluoromethanesulfonic acid, glycans, mass spectrometry, biosynthesis

**Ad Hoc Peer Reviewer** ⓘ Ashu Sharma, University at Buffalo, State University at Buffalo

Address correspondence to Paul D. Veith, pdv@unimelb.edu.au, or Eric C. Reynolds, e.reynolds@unimelb.edu.au.

The authors declare no conflict of interest.

*P*orphyromonas gingivalis is a Gram-negative anaerobic pathogen strongly associated with periodontitis in humans (1–5). Periodontitis, and in some cases *P. gingivalis* in particular, has been linked to an increased risk of cardiovascular diseases, certain cancers, preterm birth, rheumatoid arthritis, and dementia (6–10).

Glycosylation of *P. gingivalis* proteins has been reported in many studies and likely results from multiple glycosylation pathways (11). The most studied is the type IX secretion system (T9SS) sortase pathway, which brings together the synthesis of glycans associated with A-lipopolysaccharide (A-LPS) and its conjugation to T9SS cargo proteins via the PorU sortase (12, 13). The sortase is a transpeptidase that cleaves the C-terminal signal domain and creates a new peptide bond between the C-terminus of the matured cargo protein and the seryl component of the linking sugar (13). The linking sugar was determined to be 2-N-seryl, 3-N-acetylglucuronamide, which is assumed to be part of the A-LPS molecule (14). The lipid portion of A-LPS is then thought to be the membrane anchor that allows the cargo proteins to be tethered to the cell surface. Biosynthesis of the linking sugar was shown to involve both the Wbp pathway as well as VimA and VimE (14, 15). The glycosyltransferases shown to be specifically involved in the synthesis of A-LPS and cargo modification are WbaP, VimF, GtfC and GtfF (16). The effect of "A-LPS" modification of T9SS cargo proteins can be seen from their migration on SDS-PAGE as diffuse bands spread over a broad molecular weight (MW) typically 20–40 kDa higher than predicted from the protein sequence alone (17–20).

The most important cargo proteins are the gingipains, RgpA, RgpB, and Kgp, due to their major roles in nutrient acquisition and virulence (21). In addition to the C-terminal modification with A-LPS, the RgpA catalytic domain has also been reported to be glycosylated with a smaller glycan (22), indicating the possible presence of additional systems. Other proteins that appear to be glycosylated include BamA (PG0191), whose glycosylation was dependent on the GalE putative epimerase (23); the minor fimbrillin, Mfa1 (24) based mainly on carbohydrate staining; and PGN_0743, PGN_0876, PGN_1513, and PGN_0729 (Omp41), which were also identified based on carbohydrate staining (25). Omp41 together with its binding partner Omp40 could also be isolated by lectin-affinity purification, and a putative O-glycosylation site in Omp41 was identified at Ser[271] in the sequence [269]PVSCPECPEVTPVTK[283] involving a single hexose (26). While these studies have hinted at the presence of glycosylation in *P. gingivalis*, others have highlighted that the conserved enzymes and antibody cross-reactivity to glycans produced in other members of the *Bacteroidetes* phylum support that *P. gingivalis* possesses a functional *Bacteroidetes* phylum O-glycosylation system (27, 28).

The *Bacteroidetes* phylum O-glycosylation system was first described in *Bacteroides fragilis* (29). The glycosylation site motif in *B. fragilis* was elegantly demonstrated by site-directed mutagenesis to be (D)(S/T)(A/I/L/V/M/T) with the glycan O-linked to the Ser or Thr residue in the second position of the motif (29). The glycosylation was shown to only occur on proteins exported beyond the cytoplasm. Furthermore, deletion of an essential component of the glycosylation machinery resulted in impaired colonization of mouse intestines, demonstrating the *in vivo* importance of this glycosylation system (29). Over 1,000 glycoproteins are predicted in *B. fragilis* based on the presence of the O-glycosylation motif, and a total of 20 glycoproteins were identified (30).

Similarly, the O-glycosylation system of the closely related *Tannerella forsythia* has been reported with the identification of 13 glycopeptides from 4 glycoproteins with inferred glycosylation sites consistent with the motif found in *B. fragilis* (28). The structure of the glycan in *T. forsythia* ATCC 43037 was solved by NMR and MS methods to contain 10 sugars (28, 31). While both the *O*-oligosaccharyltransferase responsible for transferring the glycan to the protein as well as the glycosyltransferases involved in the biosynthesis of the "core" region of the glycan are unknown, the biosynthetic pathway of the "species-specific" portion of the glycan has been elucidated (31). The biosynthesis of the glycan, its variation between strains, and its roles in virulence were recently reviewed (32). Also recently, we completed a proteomics study of *T. forsythia* and identified 312 O-glycosylation sites in 145 glycoproteins and extended the glycosylation site motif to (D)(S/T)(A/I/L/V/M/T/S/G/C/F) (33).

In the present study, we demonstrate the presence of an O-glycosylation system in *P. gingivalis* for the first time with the mass spectrometry identification of 257 putative O-glycosylation sites within 145 glycoproteins together with the sequence of the

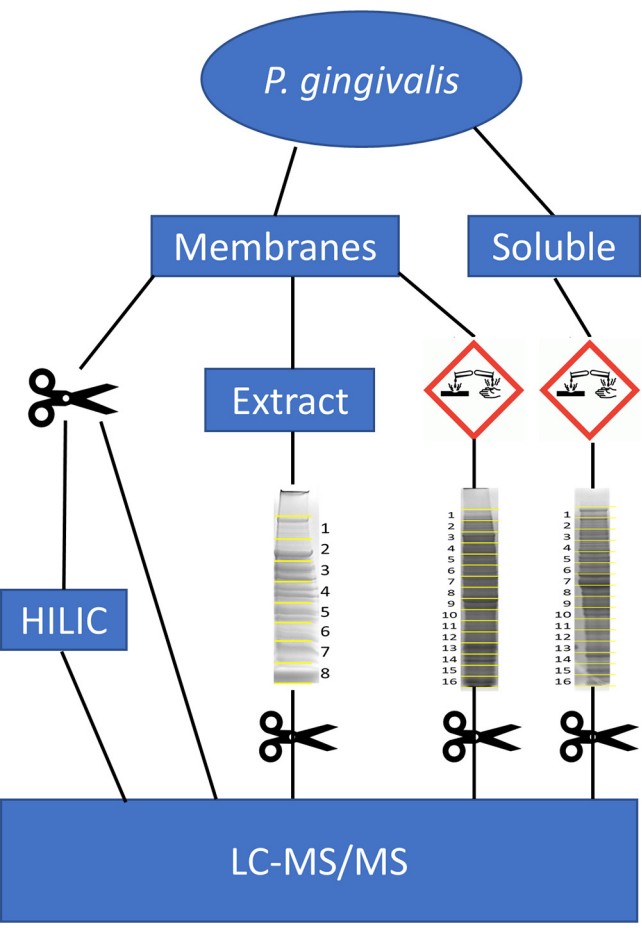

**FIG 1** Overview of sample preparation. *P. gingivalis* cells were divided into membrane and soluble fractions. A portion of the membrane fraction was digested with trypsin and the resultant peptides were analyzed by LC-MS/MS with or without HILIC enrichment. A second portion of the membrane sample was extracted with detergent, subjected to SDS-PAGE, and divided into 8 gel segments for digestion with trypsin (indicated by scissors) and LC-MS/MS. The soluble fraction as well as a third portion of the membrane sample underwent partial deglycosylation in trifluoromethanesulfonic acid (indicated by the hazard symbol) followed by SDS-PAGE separation into 16 gel segments, trypsin digestion and LC-MS/MS.

glycans and the identification of two glycosyltransferases required for the glycan synthesis.

## RESULTS

**Overall identification data.** *P. gingivalis* was fractionated, and samples with or without partial deglycosylation were digested with trypsin and analyzed by LC-MS/MS as outlined (Fig. 1). These analyses described in detail below ultimately led to the identification of 257 O-glycosylation sites within 145 glycoproteins (Tables S2 and S3 in the supplemental material).

**Survey of glycoforms.** To investigate the anticipated O-glycoproteome of *P. gingivalis*, a total membrane sample was prepared, digested with trypsin, and analyzed by LC-MS/MS with two FAIMS settings and multiple fragmentation modes. The data were searched with Byonic using a wildcard setting to enable modified peptides of any Δmass to be identified. A plot showing the number of identified peptides for frequently observed Δmass values exhibited several clusters of potential glycopeptides differing by 1 Da units (Fig. 2). The most frequent Δmass values observed were the 1436 Da and 1394 Da clusters representing 242 peptides (5.3%) (Fig. 2). Other abundant values were 128.10 and 156.10, which could be attributed to Lys and Arg respectively, presumably

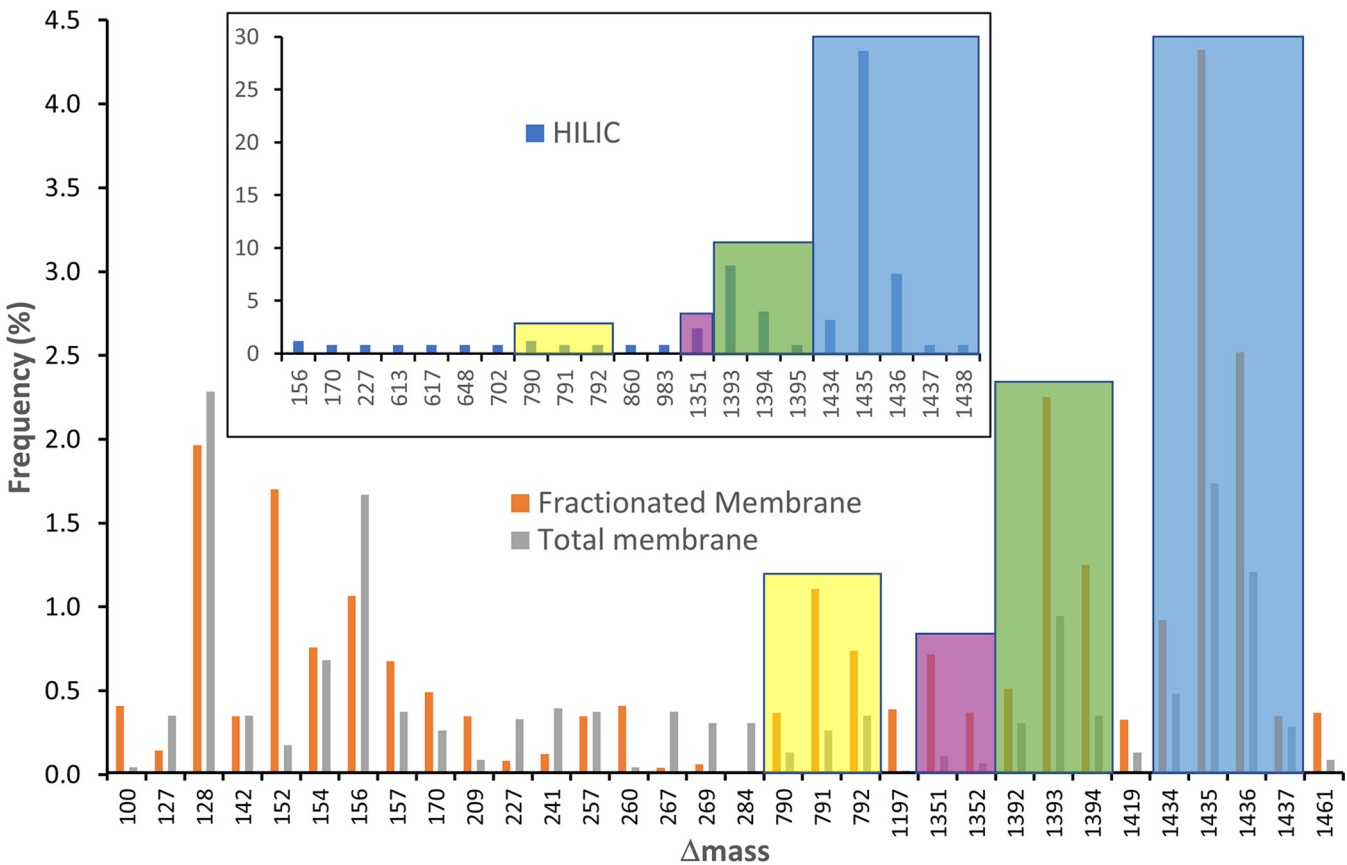

**FIG 2** Detection of glycoform clusters using Byonic wildcard. Total membrane, fractionated membrane, and HILIC-enriched samples (Fig. 1) were analyzed by LC-MS/MS and the data searched using Byonic in wildcard mode. The search results were binned into integer Δmass values, and then the frequency of each value was plotted. Only Δmass values with a frequency > 0.3% are shown.

due to the misassignment of peptides with two adjacent tryptic cleavage sites. Additional clusters were observed with Δmass values around 792 Da and 1352 Da.

A preliminary look at the MS/MS spectra of the identified peptides with potential modifications of ∼1352, ∼1394, and ∼1436 Da indicated they were glycopeptides due to the presence of sugar fragmentation and the HexNAc oxonium ion at *m/z* 204.086 (see below). Furthermore, nearly all of the peptides contained the putative Bacteroidetes O-glycosylation motif of (D)(S/T)(X) (see below), suggesting that these modifications may be the major O-glycans of *P. gingivalis*. Peptides of the 792-cluster tended to have the same O-glycosylation motif but lacked the HexNAc oxonium ion. The glycosylated proteins appeared to be mostly lipoproteins and proteins associated with the inner membrane (IM). Therefore, to enrich for these proteins and allow a deeper survey of the O-glycoproteome, the total membrane sample was extracted with a low concentration of detergent and fractionated by SDS-PAGE (Fig. 1). Trypsin digestion of the gel segments followed by the same MS analysis approach yielded 592 peptides in the 1394/1436 clusters representing 12.1% of the total (Fig. 2). A third survey for the presence of glycopeptides was conducted by performing HILIC enrichment on the total membrane digest sample. As could be expected, the 1394/1436 clusters were greatly enriched, and the presence of potential glycopeptides with Δmass values of ∼792 Da were supported (Fig. 2, inset).

To further investigate the clusters, well defined MS1-level spectra of up to 10 peptides exhibiting each Δmass value were inspected (Table S4). In total, the monoisotopic mass was recorded for 64 spectra. In all cases, the Δmass value was calculated to be 1436.43, 1394.42, 1352.42, or 792.25, and these values were assumed to correspond to the major O-glycans present. The data were then searched again using both Byonic and Mascot with these four Δmass values used as variable modifications specific to Ser

or Thr. The searches proved successful in identifying a larger number of glycosylated peptides including almost all of the expected cluster peptides from the Byonic wildcard searches confirming the presence of just four major glycans. In total, 416 different glycopeptide forms were putatively identified with a Byonic score greater than 200 (Table S5). Only 6 of these did not contain a putative Bacteroidetes O-glycosylation motif, and of these, none could be confirmed to be reliable, usually with the incorrect charge state being assigned to the precursor. Therefore, in downstream analyses, only peptides with a putative Bacteroidetes O-glycosylation motif were considered.

**Identification of the glycan sequence.** The glycan sequence was determined by ion trap resonance-based collision-induced dissociation (CID) spectra of glycopeptides containing the four glycans and confirmed by TFMS cleavage. The proposed sequence of the largest glycan with a Δmass of 1436 is HexNAc-HexNAc(P-Gro-[Ac]$_2$)-dHex-Hex-HexA-Hex(dHex). The glycans of Δmass 1436, 1394, and 1352 differed by 42.01 Da, matching an acetyl group (C$_2$H$_2$O). The CID fragmentation patterns were the same for these glycans, with the 42-Da differences being accounted for in the glycerol moiety (Fig. 3A to C). Hence, the three glycans are proposed to have either glycerol phosphate, acetyl glycerol phosphate, or diacetyl glycerol phosphate. Accurate mass data were obtained for these glycerol phosphate moieties from oxonium ions in the high resolution HCD scans. In addition to the oxonium ion for HexNAc at *m/z* 204.086, ions specific to the three glycans were observed at 442.111 (HexNAc-P-Gro-[Ac]$_2$), 400.100 (HexNAc-P-Gro-Ac), and 358.089 (HexNAc-P-Gro). The only glycoform observed with a different monosaccharide sequence was the glycan with Δmass 792, comprising a pentasaccharide of sequence dHex-Hex-HexA-Hex(dHex) (Fig. 3D). Since for the intact glycans, CID was only activated for peptides containing HexNAc, the CID spectra for Δmass 792 peptides were obtained from the acid-cleaved samples.

Since the major glycans were large, the total membrane and soluble fractions were partially deglycosylated with TFMS in an attempt to truncate the glycans. The deglycosylated samples were then fractionated by SDS-PAGE, analyzed by LC-MS/MS and initially searched with Byonic using the wildcard setting. The most frequent Δmass values observed clustered around 499/500 Da, and inspection of the corresponding peptide sequences indicated that they overlapped extensively with the peptides modified with the intact glycans and also contained the same Bacteroidetes O-glycosylation motif (Table S2). Examination of the MS1-level spectra demonstrated that the true Δmass value was 500.14, corresponding to the Hex-HexA-Hex portion of the intact glycans. The data were searched again with modifications defined in the database for each portion of the intact glycan. TFMS-cleaved glycopeptides were found with Δmass values of 162, 338, 484, 500, 646, and 792 Da matching to various portions of the first five sugars (Table 1). Acid cleavage between the terminal HexNAc residues was not detected. Acid cleaved products with Δmass 1290 Da and 1248 Da were additionally observed corresponding to full-length glycan (1436 or 1394 Da) minus one dHex residue. The most frequent Δmass observed was 500 Da, indicating the strongly preferred cleavage of dHex residues (Table 1).

To enable the characterization of the glycans of acid-cleaved glycopeptides, targeted CID spectra were acquired with the aid of inclusion lists. CID of the Δmass 1290 glycopeptides resulted in a simple, linear fragmentation pathway that matched to the full-length glycan minus the dHex side branch (compare Fig. 4B with Fig. 4A). The +484 Da fragment (Y$_2$ in Fig. 4A) was specific to the full-length glycan, suggesting the dHex side branch is located on one of the first two sugars. Its location on the reducing Hex residue was confirmed by analysis of mutants (see below). CID of the smaller acid-cleaved glycans of Δmass 646 Da (Fig. 4C), Δmass 500 Da (Fig. 4D), Δmass 338 Da (Fig. 4E), and Δmass 162 Da (Fig. 4F) produced the expected fragment ions, helping to confirm the assigned structure.

**Localization of glycosylation sites.** All peptides that were confirmed to be glycosylated exhibited an extended Bacteroidetes O-glycosylation motif of (D)(S/T)(A/I/L/V/M/T/S/G/F/C), the same as found in *T. forsythia* (33) (Table S2). Since many of these peptides did not have Asn in their sequence or any additional Ser or Thr residues, the

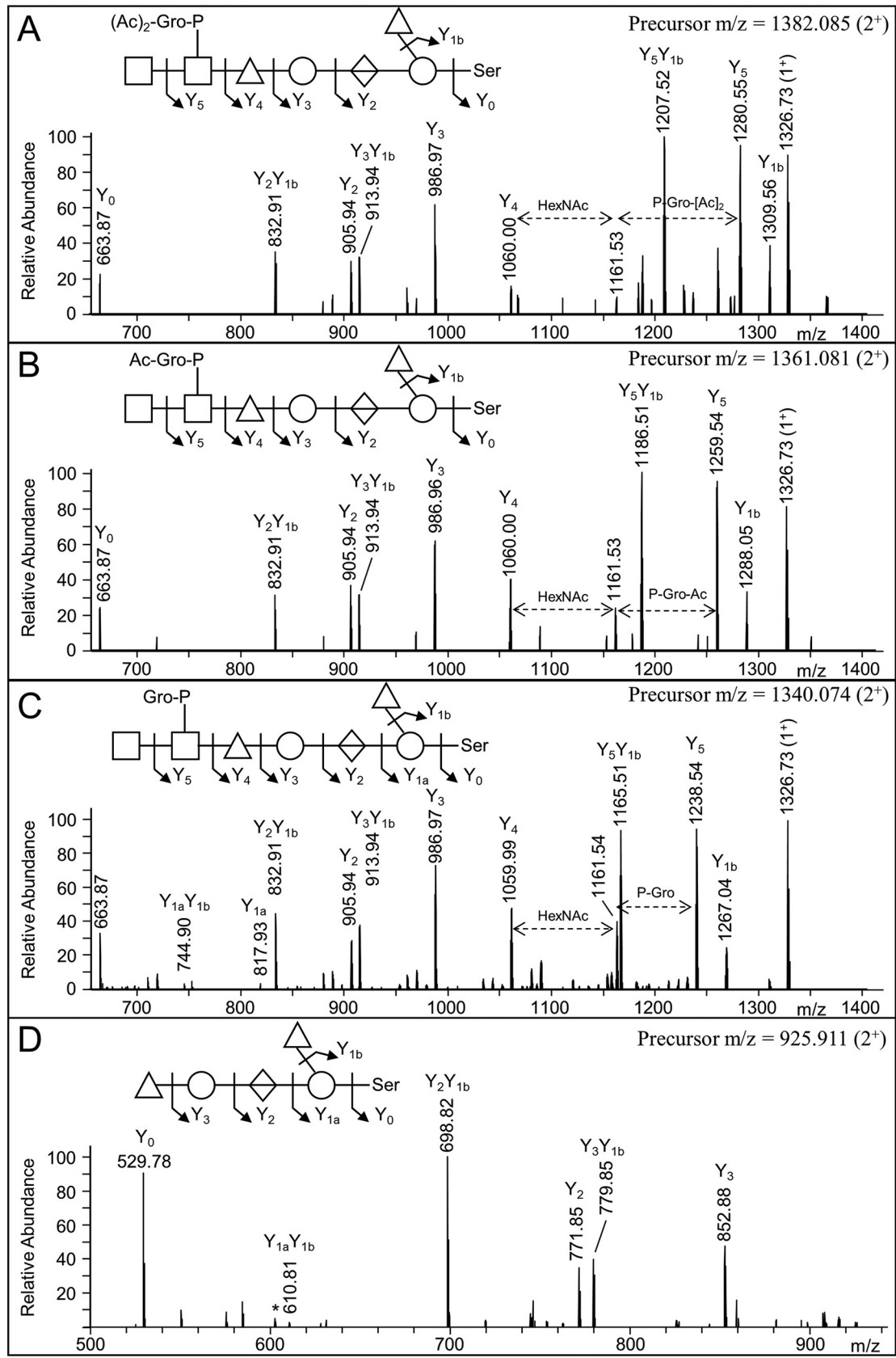

**FIG 3** CID spectra of tryptic peptides modified with the four detected glycoforms. Each CID spectrum corresponds to the peptide FTFIKDSIVEK from the protein PGN_1544 with Δmass 1436 Da (A), Δmass 1394 Da (B), Δmass 1352 Da (C), and Δmass

**TABLE 1** Frequency of acid-cleaved glycans and residual intact glycans

| Δmass | 162 | 338 | 484 | 500 | 646 | 792 | 1248 | 1290 | 1352 | 1394 | 1436 |
|---|---|---|---|---|---|---|---|---|---|---|---|
| Freq. | 36 | 82 | 9 | 265 | 41 | 19 | 10 | 13 | 12 | 22 | 22 |

glycosylation motifs were the most likely modification sites. Further evidence of modification at these sites was obtained from the "Δmod" scores, which provide the Byonic score of the top matching site minus the second-best matching site. In this study, since the modifications were defined as being linked to Ser or Thr, the score only differentiates between potential Thr and Ser sites. Generally speaking, the ETD or EThcD spectra of glycopeptides gave the best site localization data because in ETD, fragmentation occurs primarily along the peptide backbone, leaving the glycan intact. This mode of fragmentation, however, is generally only optimal for ions of high charge density, ions $<$ 800 $m/z$ and charge state $>$ +3 (34). However, occasionally, the CID spectra produced excellent localization data for doubly-charged glycopeptides, particularly after truncation of the glycans with TFMS. For both intact glycopeptides and acid-truncated glycopeptides, the proportion of glycans assigned to the Bacteroidetes O-glycosylation motif increased with the ΔMod score (Table 2). ΔMod scores above 30 always assigned the site to the motif, providing confidence that glycosylation is indeed occurring at the motif. Higher ΔMod scores were generally obtained for the acid-cleaved glycopeptides (Table 2).

While the ΔMod scores strongly support glycosylation occurring at the Bacteroidetes O-glycosylation motif, they don't show the exact modification site. To show this more directly, four spectra are provided that show the exact location of the glycan within the motifs DSI, DTV, DST, and DTL, respectively (Fig. 5A to D). In each case, the peptide shown is modified with the most abundant of the acid-cleaved glycans (Δmass = 500 Da). The first three spectra shown are EThcD spectra where fragmentation of the glycan is not evident, and series of c and or z ions demonstrate the modification site (Fig. 5A–C). The fourth is a CID spectrum (Fig. 5D) where y-ions were observed for both the glycan-cleaved peptide fragments (~y3 - ~y11) as well as the peptide fragments that retained the glycan and reveal the modification site (y3, y5–y7).

In addition to detecting glycosylated peptides, the corresponding nonglycosylated peptides were also identified for some sites. Out of the 257 identified sites, only 27 were also found in unmodified form (Table S2). This indicates that on the whole, the level of site occupancy was very high.

**Biosynthesis of the glycan.** To explore the biosynthesis of the glycan, a panel of mutants was tested by Western blotting of whole cell lysates using antibodies against the Mfa2 protein (PGN_0288) identified to have three glycosylation sites (Table S2). Of the 10 glycosyltransferase mutants plus the *waaL* ligase and a *wzx* flippase, only the PGN_1134-1135 double mutant showed an obvious shift in the MW of Mfa2 (Fig. S1), suggesting that one or both of these genes is essential for complete O-glycosylation. To determine the individual contributions of PGN_1134 and PGN_1135, single mutants were created and analyzed by Western blotting. Whole cell lysates of the PGN_1135 mutant produced similar results to the double mutant; however, PGN_1134 produced a smaller shift in the MW of Mfa2 (Fig. 6A). To ensure that the glycosylation defects were specifically due to the targeted genes, the individual mutants were complemented. The complemented strains appeared to produce Mfa2 at the same MW as the wild type (Fig. 6B), confirming that PGN_1134 and PGN_1135 are required for the normal O-glycosylation of Mfa2.

**FIG 3** Legend (Continued)

792 Da (D). The first three spectra (A–C) are from the fractionated membrane sample (gel segment #7), and the CID spectra were triggered by the presence of HexNAc (*m/z* 204.09) in the corresponding HCD spectra. The fourth CID spectrum (D) was taken from the deglycosylated data set (targeted analysis) since it lacks HexNAc. For all spectra, the labeled ions are 2+ unless shown otherwise. Sugar symbols follow the SNFG as described in Materials and Methods. *, Indicates ions that may be produced by dHex scrambling.

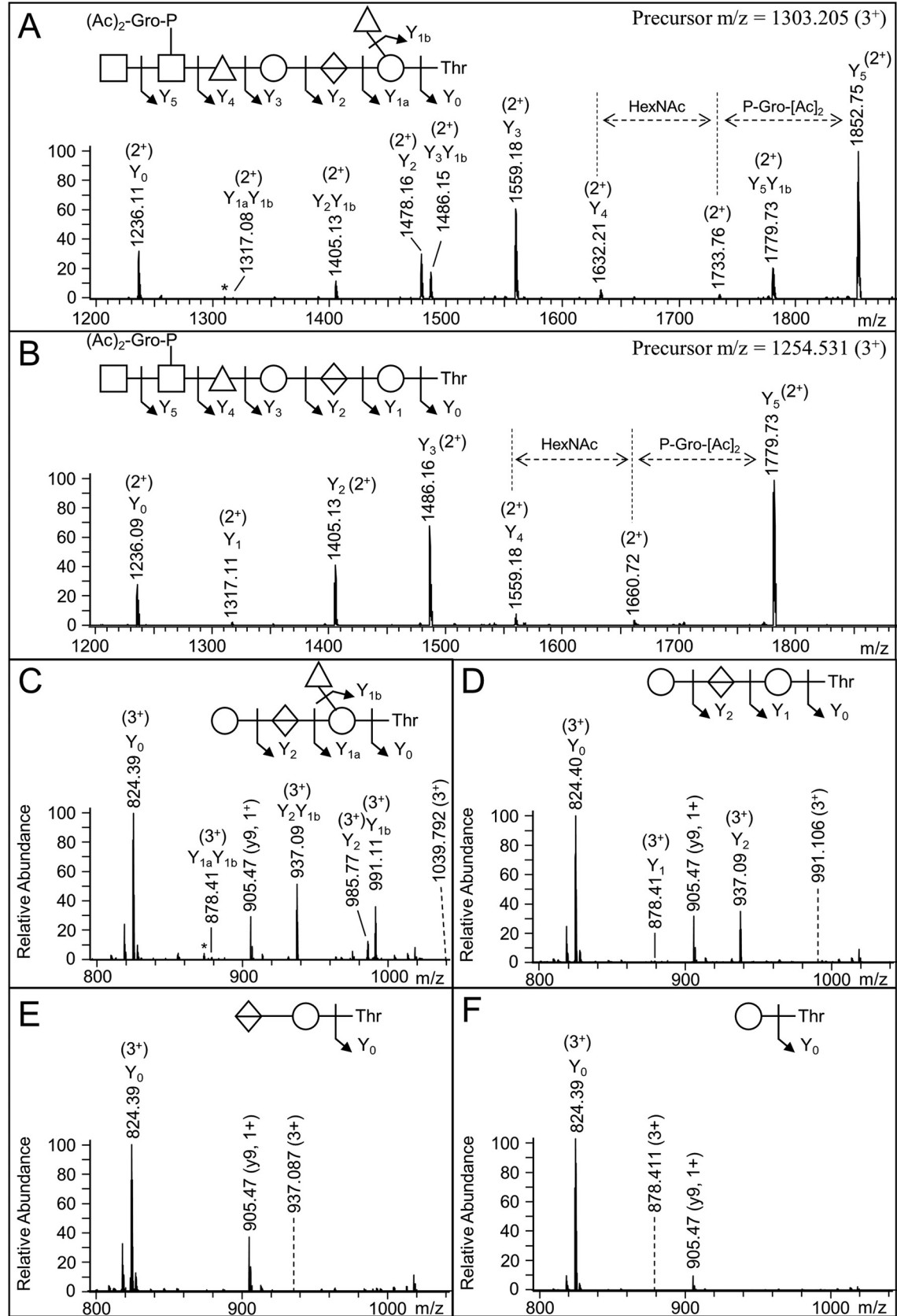

**FIG 4** CID spectra of partially deglycosylated peptides. Each CID spectrum corresponds to the peptide GGKEDGSGSTNSGAFTISGDTVSLAR from the protein PGN_1037 with either Δmass 1436 Da (full length, A), Δmass 1290 Da (B), Δmass 646 Da (C), Δmass 500 Da (D), Δmass

**TABLE 2** Localization of glycosylation sites to the O-glycosylation motif using ΔMod Scores

| Sample type | ΔMod score >15 | ΔMod score >30 | ΔMod score (Av) |
|---|---|---|---|
| Intact | 27/34 | 14/14 | 44 |
| Acid-cleaved | 26/31 | 19/19 | 62 |

To verify that these genes are also required for the O-glycosylation of other proteins and to identify the steps in glycan synthesis blocked by these genes, proteins from whole cell lysates of each mutant strain were separated by SDS-PAGE, in-gel digested, and analyzed by LC-MS/MS. The double mutant and PGN_1135 mutant exhibited similar results, with the largest glycan having a Δmass of 792, which was observed for 124 and 155 nonredundant peptides respectively (Table 3). The next most common glycan observed exhibited a Δmass of 308, with ~30 peptides suggesting a partial blockage at that step. The largest glycan observed in the PGN_1134 mutant exhibited a Δmass of 995 on 87 peptides; however, a Δmass of 792 was most common, with 157 peptides identified. Inspection of the MS/MS data revealed that peptides with a Δmass of 792 in all mutants fragmented the same as that shown in Fig. 3D suggesting the same pentasaccharide structure. The MS/MS data for peptides with Δmass 308 helpfully confirmed the location of the branched deoxyhexose on the reducing hexose (Fig. 7A). The MS/MS data for peptides with Δmass 995 were consistent with the same pentasaccharide plus a HexNAc residue (Fig. 7B). Assuming the largest glycans observed in each mutant are the direct substrates of the corresponding enzymes, PGN_1135 is inferred to transfer a HexNAc residue to the pentasaccharide and PGN_1134 is inferred to transfer the terminal HexNAc. Since phosphoglycerol substituents were not identified, it is likely this moiety is transferred at a later step (Fig. 7C).

**Glycan heterogeneity.** Further Mascot searches of the intact wild-type samples were conducted to determine whether the phosphoglycerol addition was stoichiometric. Small numbers of glycopeptides with Δmass values of 995.33 Da and 1198.41 Da corresponding to the inclusion of one or both HexNAc residues without phosphoglycerol were identified. The abundance (peak intensity) of these glycopeptides was typically <1% relative to those with phosphoglycerol, suggesting that the modification was nearly stoichiometric. The HPLC retention times of these glycopeptides were always less than their three counterpart glycopeptides substituted with variously acetylated glycerophosphate, confirming genuine sample heterogeneity rather than fragmentation within the mass spectrometer (Fig. S2). Furthermore, acetylation consistently resulted in a slight increase in retention time, again confirming the genuine heterogeneity of acetylation levels (Fig. S2).

**The proteins identified.** The putative localizations of the 145 glycoproteins identified were inner membrane (IM, 48), lipoproteins (42), periplasm (28), outer membrane (OM, 15), T9SS cargo (4), and uncertain (8) (Table S3). Due to the familiarity of the authors with the OM proteome (20, 35), it was immediately apparent that the most abundant OM-associated proteins were absent from the list of identified glycoproteins. To explore this observation further, the theoretical OM proteome and lists of predicted lipoproteins, periplasmic proteins, and IM proteins (36) were examined in detail to determine patterns between abundance, various protein categories, predicted glycosylation sites, and detected glycosylation (Table S6). For the 10 most abundant OM proteins, only three (30%) had predicted O-glycosylation sites (Fig. 8); however, none were identified to be glycosylated, and furthermore the unmodified forms of the peptides bearing the three sites were identified, suggesting that the predicted sites may not be modified at all. In contrast, 80% of the abundant periplasmic proteins, and 100% of the

**FIG 4** Legend (Continued)

338 Da (E), or Δmass 162 Da (F). Besides the spectrum corresponding to a full-length glycan (A), the spectra were taken from the deglycosylated data set (targeted analysis). The precursor *m/z* values are provided in the top right of each spectrum or indicated by a dashed line within the spectra. Sugar symbols follow the SNFG as described in Materials and Methods. *, Indicates ions that may be produced by dHex scrambling.

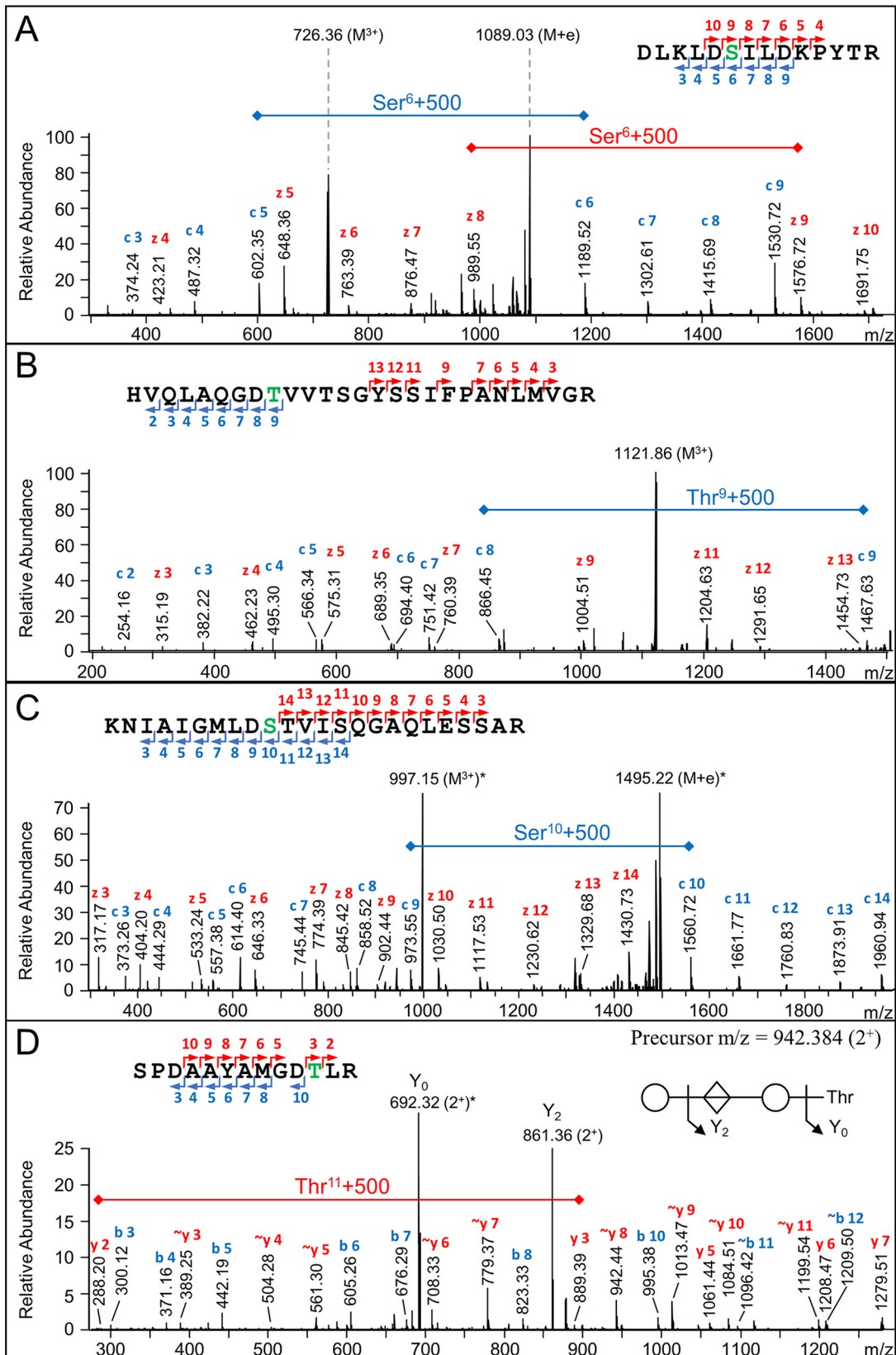

**FIG 5** Localization of glycosylation sites. EtHCD spectra (A–C) and CID spectrum (D) showing localization of the residual glycan (Δmass = 500 Da) at the Ser or Thr residue indicated in green within four different peptides. All spectra are from the deglycosylated data set (targeted analysis). All ions are $1^+$ unless otherwise indicated. For the EtHCD spectra, the precursor is indicated in the spectra by the letter M. Intense ions indicated with an * had a relative abundance of 100%. For the CID spectrum (D), cleavage of the glycan is also indicated.

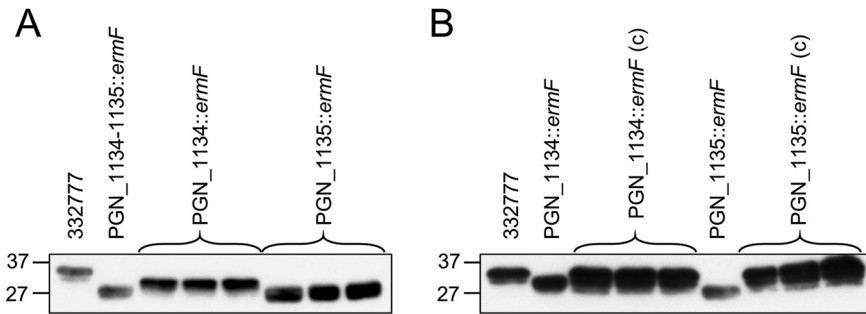

**FIG 6** Western blot of PGN_1134 and PGN_1135 mutants. Whole cell lysates of each strain were analyzed by Western blotting using antibodies against Mfa2. (A) One clone of the double mutant and three different clones for each single mutant were tested as shown. (B) The single mutants were compared to their respective complemented strains (3 clones each). PGN 1134::ermF (c) = PGN_1134::ermF/pTCB– catalase pro-PGN_1134. PGN_1135::ermF (c) = PGN_1135::ermF/pTCB–catalase pro-PGN_1135.

abundant IM proteins and non-localized lipoproteins, were predicted to be glycosylated, with a total of 13, 38, and 42 predicted sites, respectively (Fig. 8). The overall proportion of proteins glycosylated was substantially lower (55%) for proteins localized to the OM compared to the other locales, which ranged from 71% to 80% (Fig. 8). Together, the data suggest that the O-glycosylation of OM-associated proteins is non-preferred, particularly if they are abundant.

## DISCUSSION

This is the first detailed report of O-glycosylation in *P. gingivalis* and the first to directly identify O-linked glycans. This makes O-glycosylation the second protein glycosylation system to be described for this species, after the T9SS-related system, which glycosylates the mature C-terminus of cargo proteins via a peptide linkage (13, 14). The glycan was shown to be a heptasaccharide substituted with acetyl glycerol phosphate, which has not been observed in other bacterial glycans to our knowledge. We identified 145 O-glycosylated proteins, which is 40% of the total number (358) predicted for *P. gingivalis* ATCC 33277 based on the presence of the O-glycosylation motif (27).

The first six monosaccharide classes of the heptasaccharide are almost identical to the first six in the putative *B. fragilis* O-glycan (37). The only detected differences are in the substitutions of acetyl glycerol phosphate or O-methyl (Fig. 9A and B). Furthermore, the first four sugars of the *B. fragilis* O-glycan may be the same as that of *Elizabethkingia meningoseptica* (Fig. 9C) (38). The most different of the four *Bacteroidetes* O-glycans is that produced by *T. forsythia*, which has branches on every monosaccharide, but it still shares a similar trisaccharide unit of HexA-Hex(dHex) (Fig. 9D). The dHex branch in *E. meningoseptica* is O-methylrhamnose, and the dHex branch in *B. fragilis* was inferred to be the same (27). While the dHex branch in *T. forsythia* was reported to be fucose, there is some uncertainty of this because it was originally identified by NMR and thought to branch from the Me-Gal residue (28). Its position branching from the reducing-end Gal was deduced by MS methods (31) and therefore could in theory be a different deoxyhexose. The reducing end sugar is Gal in *T. forsythia* but Man in *E. meningoseptica*, so despite commonalities in the first 3–4 sugars, there are also differences. Despite these differences, an antibody that recognizes the "core glycan" of *B. fragilis* also cross-reacts with glycoproteins from all four species and many other species across all major classes within the *Bacteroidetes* phylum (27).

Of these four *Bacteroidetes* O-glycans, the biosynthesis has been best studied in *T. forsythia* where the roles of five glycosyltransferases have been elucidated (31).

**TABLE 3** Glycan Δmass frequency observed in mutants lacking PGN_1134 or PGN_1135

| Δmass | 162 | 308 | 338 | 484 | 500 | 646 | 792 | 995 | 1394 | 1436 |
|---|---|---|---|---|---|---|---|---|---|---|
| PGN_1135 | 5 | 33 | 0 | 2 | 1 | 4 | 155 | 0 | 0 | 0 |
| PGN_1134 | 6 | 15 | 2 | 5 | 0 | 10 | 157 | 87 | 0 | 3 |
| PGN_1134-1135 | 4 | 29 | 0 | 4 | 0 | 13 | 124 | 0 | 0 | 0 |

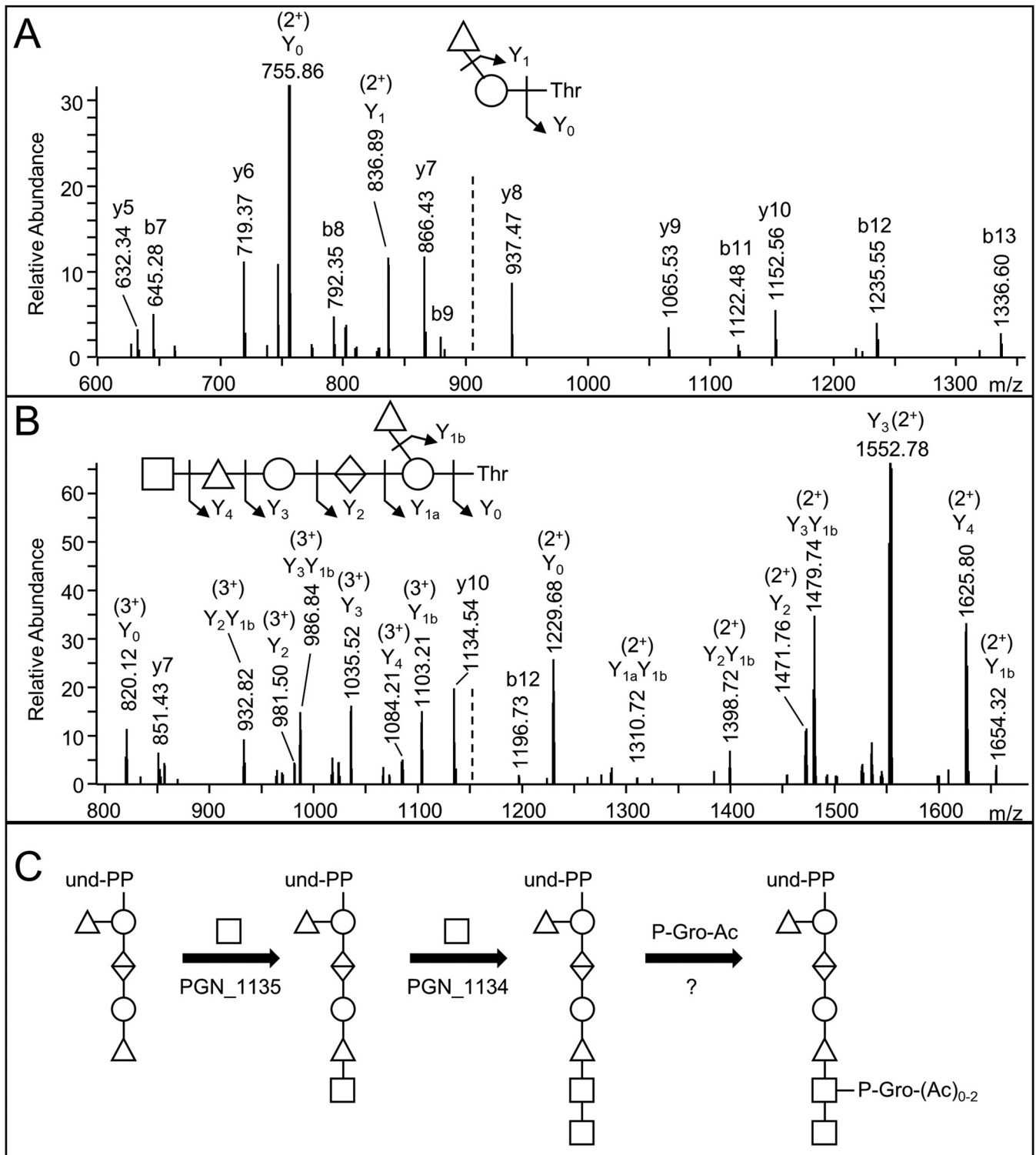

**FIG 7** Biosynthesis of the *P. gingivalis* O-glycan. (A) CID spectrum of ADTASQAFSNEITR +308 (*m/z* = 909.92, 2+) from the PGN_1473 protein in the PGN_1135 mutant. (B) CID spectrum of AALLKDTVSIALKPADAQAFMQR +995 (*m/z* 1151.90, 3+) from the PGN_0742 protein in the PGN_1134 mutant. The b- and y-ions shown are singly charged while the charges of Y-ions are indicated. (C) Proposed reactions catalyzed by the PGN_1135 and PGN_1134 glycosyltransferases.

BLAST searches of these enzymes demonstrated the existence of close orthologs for GtfE and GtfL in both *P. gingivalis* and *B. fragilis* but not *E. meningoseptica* (Table 4). In *T. forsythia*, GtfE was proposed to transfer the branching dHex residue; however, it appears more likely that it transfers the internal Fuc residue (Fig. 9D) since the

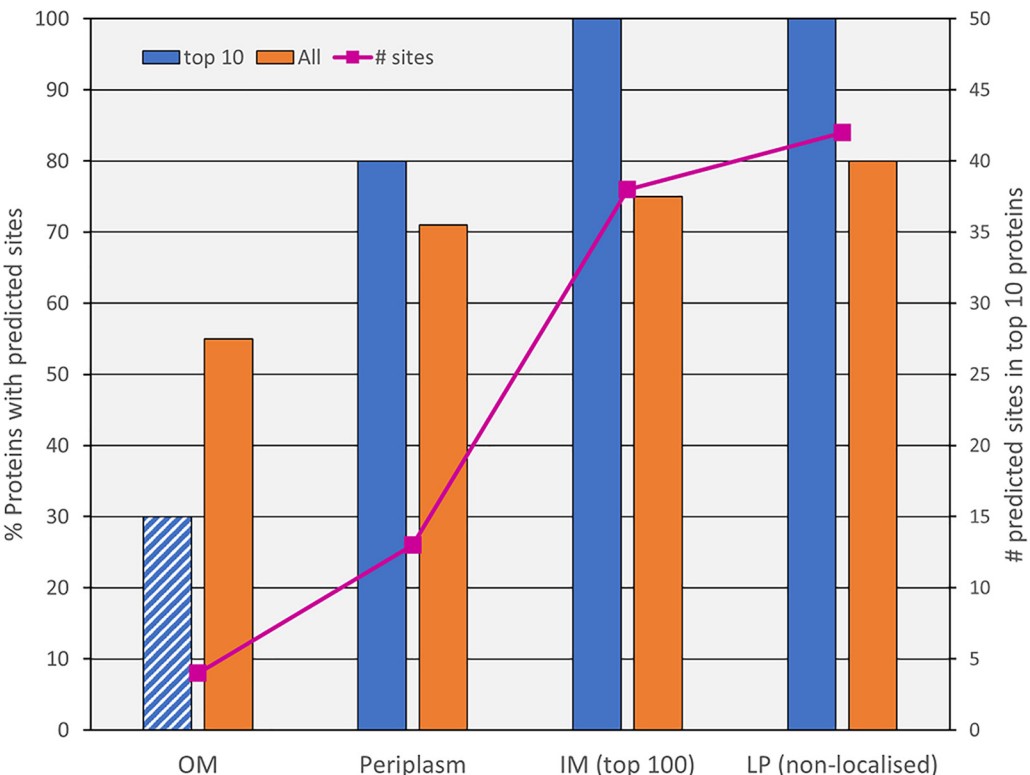

**FIG 8** The O-glycosylation system is biased against abundant OM proteins. The proportion of proteins with predicted O-glycosylation sites is plotted for the 10 most abundant proteins (blue) and all proteins (orange) for each localization category. The first column has a diagonal pattern to indicate that these three proteins may not be glycosylated at all since only the unmodified sites were identified. The line graph (pink) relates to the right-hand axis and shows the number of predicted sites for the 10 most abundant proteins in each category. The complete data are provided in Table S6.

branching dHex is an optional side-branch that is not present in some native glycans and yet the *gtfE* mutant produces a pentasaccharide ($Gal_1Dig_1dHex_1GlcA_1Xyl_1$) as the largest glycan detected (31). If the optional branching dHex residue was transferred by GtfE, full-length glycans (minus dHex) should have been observed in the mutant. This interpretation also fits better with the data as a whole so that the biosynthetic locus encodes for glycosyltransferases transferring adjacent sugars in the outer core (Fig. 9D). We propose that the GtfE orthologs, BF_4306 and PGN_0777, may transfer the putative internal Fuc residue in their respective glycans (Fig. 9A and B). This putative Fuc is strongly inferred in *B. fragilis* (27), and proposed in *P. gingivalis* by homology of structure and biosynthesis (Fig. 9). In *T. forsythia*, GtfL transfers the Me-Gal residue to the internal Fuc since in the gtfL mutant, the largest glycan observed was the hexasaccharide (31). Similarly, in the PGN_1135 mutant, the largest glycan was the pentasaccharide ($\Delta$mass = 792, Table 3), consistent with transferring the first HexNAc residue (Fig. 7C), which may be GalNAc since the homology to GtfL is so strong (Table 4). Similarly, in the PGN_1134 mutant, a hexasaccharide was observed consistent with PGN_1134 transferring the terminal HexNAc (Fig. 7C, Fig. 9A).

Interestingly, in *B. fragilis* mutants lacking the whole "LFG" glycosylation locus spanning BF_4298-4306 or just lacking BF_4306, or when GDP-fucose production was blocked, only a residual O-glycan disaccharide comprising (Me)dHex-Hex was reported (27). However, it is not clear whether larger glycans may also have been present. In our own data (Table 3), the corresponding disaccharide (dHex-Hex) of $\Delta$mass 308 was also observed in the tested mutants, suggesting there may be a significant additional blockage at the step of transferring the third sugar. This blockage appears to support the model where the disaccharide core glycan is transferred to the protein first, and then the

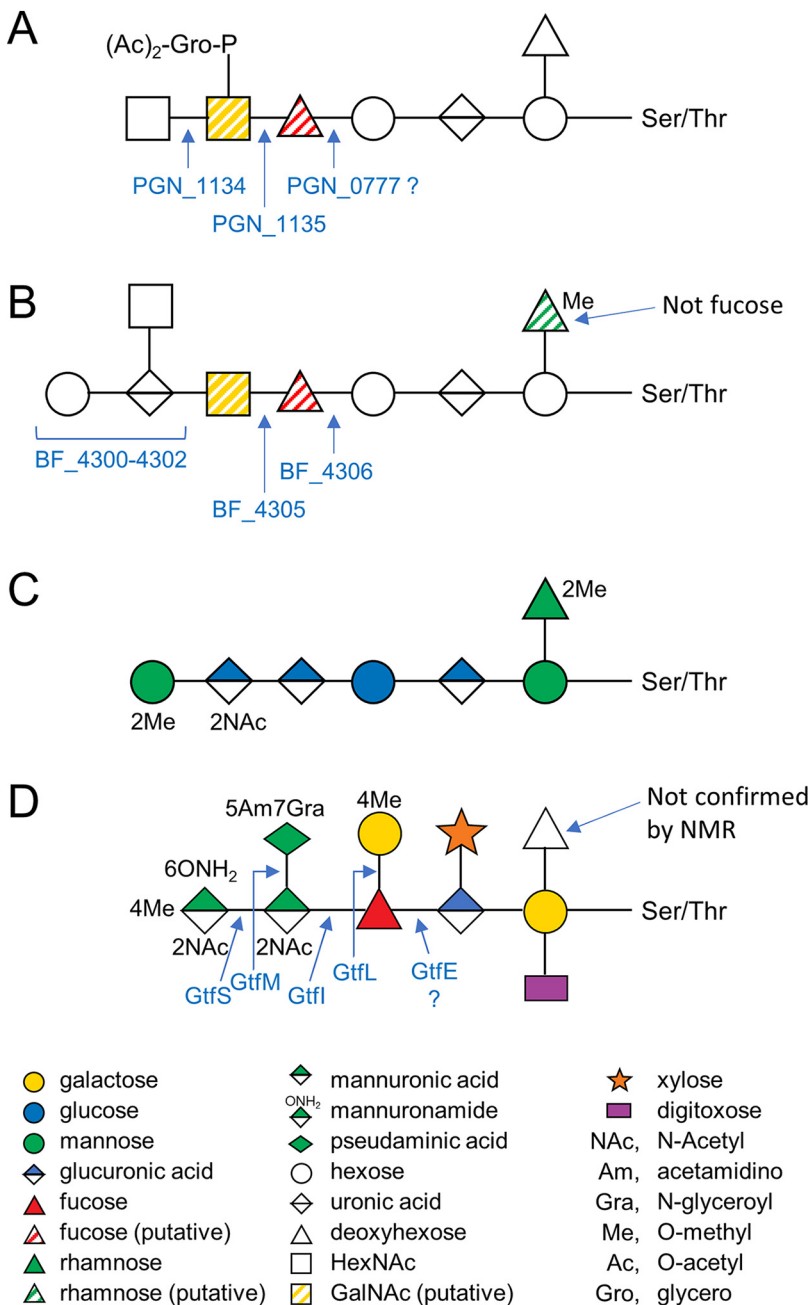

**FIG 9** Comparison of O-glycan structure and biosynthesis among *Bacteroidetes*. O-glycans from (A) *P. gingivalis* ATCC 33277 (this study), (B) *B. fragilis* (37). (C) *E. meningoseptica* (38) and (D) *T. forsythia* (31). Glycosyltransferases predicted or shown to be involved in the biosynthesis are shown. Note that striped sugar symbols are not part of the SNFG but are used here to show the predicted isomeric form of the sugar.

outer glycan is subsequently ligated to the core (27). In *P. gingivalis*, the smallest native glycan observed was the pentasaccharide (Δmass 792), suggesting that the outer core must have a minimum of dHexA-Hex-HexA to be recognized by the ligase and be transferred to the core. The ligase may have a preference for transferring a full-length outer core such that any deletion of the outer core glycosyltransferases leads to some proportion of the glycosylation sites being occupied by the core disaccharide only.

Our data confirm some of the previously identified glycoproteins of *P. gingivalis*. Omp85, now more commonly known as BamA, is an outer membrane protein that catalyzes the insertion of β-barrel outer membrane proteins into the OM (39). BamA was

**TABLE 4** BLAST data for selected glycosyltransferases[a]

| *P. gingivalis* ATCC 33277 | *T. forsythia* ATCC 43037 | *B. fragilis* NCTC 9343 | *E. meningoseptica* |
|---|---|---|---|
| PGN_0777 | Tanf_01305 GtfE 53% ID | BF4306 50% ID | 40% ID, 32% Cov |
| PGN_1135 | Tanf_01300 GtfL 45% ID | BF4305 41% ID | 27% ID, 49% Cov |
| PGN_1134 | – | 28% ID, 65% Cov | – |
| – | Tanf_01290 GtfI | BF4301 51% ID | ND |
| – | Tanf_01260 GtfM | – | ND |
| – | Tanf_01245 GtfS | – | ND |
| – | - | BF4300 | ND |
| 25% ID, 56% Cov | 27% ID, 46% Cov | BF4299 | ND |

[a]Dashes represent no significant hits. Shading represents highly significant matches over the full length of the protein. ID, identity; Cov, sequence coverage; ND, BLAST search not done.

previously suggested to be glycosylated by a reduction in MW after TFMS treatment or by mutation of the putative epimerase GalE (23). Here, we confirm that BamA (PGN_0299) is O-glycosylated at [280]Ser, and inspection of the sequence reveals two further predicted sites ([373]Ser and [31]Thr), helping to explain the significant loss in MW observed after deglycosylation. In addition to BamA, other *P. gingivalis* proteins involved in the translocation and assembly of Omps (36) were identified to be glycosylated, including associated lipoproteins and chaperones (PGN_0296 and PGN_0301), TamA (PGN_0973 and PGN_0147), and TamB (PGN_0145 and PGN_0148) (Table S3).

Of proteins previously thought to be glycosylated on the basis of carbohydrate staining (25), we established that the peptidyl-prolyl cis-trans isomerase PGN_0743 and the TPR domain protein PGN_1513, both periplasmic proteins, are O-glycosylated (Table S3); however, PGN_0876 and the very abundant Omp41 (PGN_0729) were not found to be O-glycosylated, at least not with the glycans observed in this study.

It is interesting to note than in *T. forsythia*, the very abundant surface-layer proteins, TfsA and TfsB, are highly O-glycosylated (28). Along with 19 sites in TfsB and 11 sites in TfsA, we recently identified O-glycosylation to be a major feature of *T. forsythia* T9SS cargo proteins, with 90 additional sites found in 16 cargo proteins (33). In contrast, only 7 sites from 5 *P. gingivalis* cargo proteins were identified. Importantly, the most abundant cargo proteins, the gingipains, were not identified to be O-glycosylated. Consistent with that, inspection of their sequences revealed a complete absence of the (D)(S/T)(A/I/L/V/M/T/S/C/F/G) O-glycosylation motif. Given the low specificity of the motif, and hence the relative ease of producing the motif through mutation, the lack of O-glycosylation sites in the T9SS cargo proteins suggests that their glycosylation is being selected against. A possible further explanation for this is that *P. gingivalis* has smooth LPS (contains a long repeating polysaccharide) whereas *T. forsythia* only has rough LPS (short polysaccharide). The O-glycans of T. forsythia may therefore play an important role as the dominant cell-surface polysaccharide.

O-glycosylation in *B. fragilis* is speculated to have a stabilizing function since non-glycosylated forms of the glycoproteins were not observed and site-directed mutagenesis of glycosylation sites on a particular glycoprotein appeared to destabilize it (29). It is likely that O-glycosylation in *P. gingivalis* serves a similar purpose.

The *P. gingivalis* O-glycosylation system appears to prefer proteins associated with the IM and periplasm, at both protein and glycosylation site levels. Most striking was the bias against glycosylating abundant OM-associated proteins such as the gingipains, the RagAB peptide transporter (40), and the OmpA-like proteins Omp40 (OmpA2) and Omp41 (OmpA1) (41, 42) (Table S6). Some of the abundant proteins including RagA, Omp41, PGN_1323, and PGN_0741 had predicted glycosylation sites, but only the unmodified sites were identified consistent with no or little glycosylation (Table S6). Two potential reasons for evolution selecting against the O-glycosylation of these proteins could be (i) the high cost of glycosylating abundant proteins relative to any benefit gained; (ii) adverse alteration of the bulk chemical properties of the OM locale; or (iii) to evade human serum antibody against O-glycans of oral *Bacteroidales* spp. such as

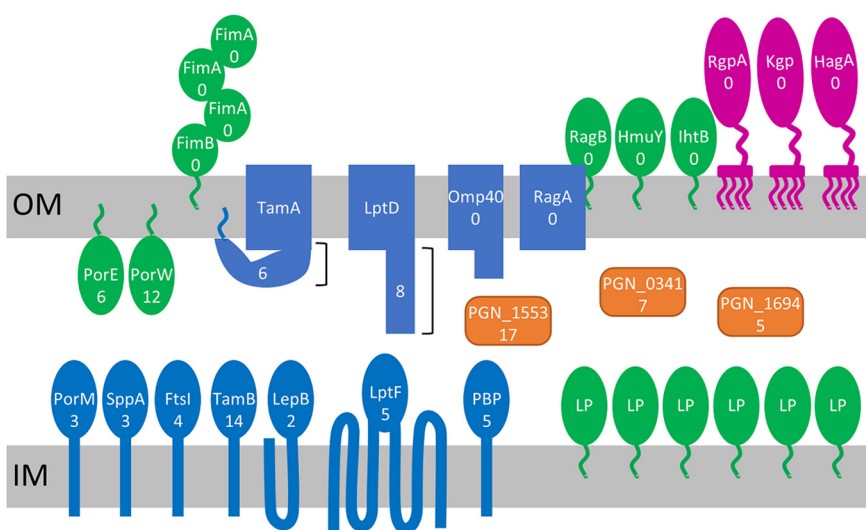

**FIG 10** Localization trends of O-glycosylated proteins. For highly abundant surface exposed proteins, there were no predicted O-glycosylation sites, whereas less abundant OM-associated proteins, particularly those with large periplasmic domains, were frequently O-glycosylated. In contrast, most proteins associated with the IM or periplasm were O-glycosylated regardless of their abundance, so long as they had a periplasmic domain. Many lipoproteins are likely to be associated with the IM, but none have been precisely localized to our knowledge; hence, they are not named in this figure. The numbers shown are the numbers of predicted O-glycosylation sites, and the two brackets near TamA and LptD indicate that the numbers only refer to the periplasmic domains. Lipoproteins are represented by green ovals with green lipid anchors; IM proteins are blue ovals with the number and topology of transmembrane helices shown; periplasmic proteins are in orange; $\beta$-barrel OM proteins are shown as blue rectangles, sometimes associated with variously shaped periplasmic domains; and T9SS cargo proteins are shown in deep pink attached to the OM via A-LPS. The locus numbers for the proteins are FimA (PGN_0180), FimB (PGN_0181), FtsI (PGN_0622), HagA (PGN_1733), HmuY (PGN_0558), IhtB (PGN_0705), Kgp (PGN_1728), LepB (PGN_1946), LptD (PGN_0884), LptF (PGN_0642), Omp40 (PGN_0728), PBP (PGN_0817), PorE (PGN_1296), PorM (PGN_1674), PorW (PGN_1877), RagA (PGN_0293), RagB (PGN_0294), RgpA (PGN_1970), SppA (PGN_0680), TamA, (PGN_0147), and TamB (PGN_0145). RagA has two predicted glycosylation sites, but both of these were identified by MS to be unmodified.

*Tannerella forsythia.* The evolutionary adaption of O-glycosylation may be relatively fast since the glycosylation motif is not very specific, allowing numerous possibilities for most proteins to evolve an O-glycosylation site through a single site mutation. If true, the affected locale is likely to be the cell surface. The T9SS cargo proteins surround the whole cells to form a surface layer and are glycosylated at the C-terminus with A-LPS (14), which may provide the desired surface chemistry or antigen presentation for *P. gingivalis* cells. Perhaps the O-glycosylation of abundant cargo proteins and other surface-exposed proteins has adverse effects on this surface chemistry or antigen presentation, leading to negative selection pressure. This theory would account for the lack of O-glycosylation found in abundant cargo proteins, OM $\beta$-barrel proteins, abundant pilins such as FimA and Mfa1, and surface-exposed lipoproteins such as RagB, HmuY, and IhtB (36). Glycosylation of low abundance surface proteins was generally limited to one or two sites and may be tolerated due to their low abundance being of little consequence to the overall surface chemistry.

The main locale of O-glycosylation was the periplasm and in particular the side closest to the IM. For the OM-associated proteins, the highest number of glycosylation sites were found or predicted in proteins with large periplasmic domains such as PorW and PorE (43, 44) as well as LptD and TamA (36) (Fig. 10). Similarly, since O-glycosylation does not occur in the cytoplasm, the IM-associated proteins found to be O-glycosylated were also mainly those with large periplasmic domains such as fusion proteins, TamB, LptF, signal peptidases, PorM, penicillin binding proteins, and cell division proteins (Fig. 10, Table S3). In summary, *P. gingivalis* has two major glycosylation systems, the T9SS-related system targeting cargo proteins localized to the cell surface, and the

O-glycosylation system that mainly targets protein domains that are localized to the periplasm.

## MATERIALS AND METHODS

**Growth of *P. gingivalis*.** Bacterial strains and plasmids created in this study as well as strains made in previous studies (16, 45, 46) are listed in Table S1. *P. gingivalis* ATCC 33277 and mutant strains were grown on solid medium containing Trypticase soy agar (40 g/L), brain heart infusion broth (5 g/L), 5% (vol/vol) lysed defibrinated horse blood, cysteine hydrochloride (0.5 g/L), and menadione (5 $\mu$g/mL) (TSBHI agar) or in tryptic soy (TS)-enriched brain heart infusion broth (TSBHI) (25 g/L tryptic soy, 30 g/L BHI broth) supplemented with 0.5 mg/mL cysteine, 5 $\mu$g/mL hemin, and 5 $\mu$g/mL menadione, both under anaerobic conditions (80% $N_2$, 10% $H_2$, and 10% $CO_2$) at 37°C. Luria-Bertani (LB) broth and LB agar plates were used for growth of *Escherichia coli* strains. Antibiotics were used at the following concentrations: ampicillin (Ap; 100 $\mu$g/mL for *E. coli*, 10 $\mu$g/mL for *P. gingivalis*) and erythromycin (Em; 10 $\mu$g/mL for *P. gingivalis*).

**Construction of the PGN_1010 mutant.** The *P. gingivalis* PGN_1010 (domain of unknown function 5020) mutant was constructed by double recombination of the target gene and the introduction of *ermF*, as previously described (47). The targeting DNA was constructed as follows. The upstream and downstream regions of the PGN_1010 gene were amplified with two pairs of primers (PGN1010upFw/ PGN1010upRv; PGN1010dwFw/PGN1010dwRv), using the genome of ATCC 33277 as template. The *ermF* region in the *ermF* DNA cassette was amplified with ermF-Fw/ermF-Rv using the genome of *gtfF* (PGN_1668)::*ermF* mutant (KDP611) as template (16). Using the three purified products, further PCR was performed with a PGN1010upFw/PGN1010dwRv primer pair. The amplified PCR fragment was cloned into pUC118, yielding pUC118-PGN_1010::*ermF*. Then, pUC118-PGN_1010::*ermF* was linearized by EcoRI. The linearized DNA was introduced into *P. gingivalis* ATCC 33277 by electroporation to generate Em$^r$ transformant (KDP1106). Transformants were selected on blood agar plates containing 10 $\mu$g/mL erythromycin.

**Construction of the PGN_1134 and PGN_1135 mutants.** To construct the PGN_1134 or PGN_1135 mutants, the downstream region of the PGN_1134 gene or upstream region of the PGN_1135 gene was amplified with PGN1134dwFw/PGN1134dwRv or PGN1135upFw/PGN1135upRv primer pairs, respectively, using *P. gingivalis* ATCC 33277 genome as the DNA template, and was cloned into pUC118 (TaKaRa, Kusatsu, Japan). The downstream region and upstream region were then purified using lower PstI-SacI sites or lower SphI-BamHI sites from each cloned recombinant plasmid and then exchanged with the corresponding sites of pKD990, which is PGN_1134-1135::*ermF* in pGEM T-easy vector (16), yielding PGN_1134::*ermF* or PGN_1135::*ermF* in pGEM T-easy vector. Finally, each desired recombinant plasmid was digested with SacI and introduced into *P. gingivalis* ATCC 33277 by electroporation to generate Em$^r$ transformant, yielding PGN_1134 (KDP1107) or PGN_1135 (KDP1108) mutants. Transformants were selected on blood agar plates containing 10 $\mu$g/mL erythromycin.

**Construction of *P. gingivalis* complemented strains.** To create complementation plasmids of the PGN_1134 and PGN_1135 genes from *P. gingivalis* ATCC 33277, the coding regions were amplified using the corresponding compFw/compRev primer pairs. Each PCR product was cloned into pUC118, yielding pUC118-PGN_1134 and pUC118-PGN_1135.

The SalI-XbaI DNA fragments containing the coding region of PGN_1134 and PGN_1135 genes from pUC118-PGN_1134 and pUC118-PGN_1135 were inserted into the same sites of pBSSK-cat pro-rgpBter (46) to yield pBSSK-cat pro-PGN_1134-rgpBter and pBSSK-cat pro-PGN_1135-rgpBter. The KpnI-NotI DNA fragment of pBSSK-cat pro-PGN_1134-rgpBter and pBSSK-cat pro-PGN_1135-rgpBter was inserted into the same sites of a pTCB vector (48) to yield pTCB-PGN_1134 and pTCB- PGN_1135. The pTCB vector containing the PGN_1134 or PGN_1135 gene was introduced into *Escherichia coli* S17-1 (49) by chemical transformation. Colonies were selected on LB agar plates containing Ap (100 $\mu$g/L). The single gene mutant was mated with PGN_1134::Em$^r$ (KDP1107) or PGN_1135::Em$^r$ (KDP1108). Colonies were selected on blood agar plates containing gentamicin (50 $\mu$g/mL) and Tc (0.7 $\mu$g/mL), yielding PGN_1134::Em$^r$/ pPGN_1134+ (KDP1109) and PGN_1135::Em$^r$/pPGN_1135+ (KDP1110).

**Cell fractionation.** *P. gingivalis* was grown in broth for 4 days and harvested by centrifugation at 8,000 *g*. Cells were resuspended in 50-mM sodium acetate, pH 5.3, and lysed using a sonication probe (model CPX 750, Cole Parmer) fitted with a 6.5-mm tapered microtip. The amplitude was set to 40% and the pulser to 1 s on, 2 s off for a total of 30 min. Unlysed cells were removed by centrifugation at 8,000 *g*. The membrane fraction was separated from the soluble fraction by ultracentrifugation at 100,000 *g* for 30 min. A portion of the soluble fraction was precipitated with 13% trichloroacetic acid and washed with ice-cold acetone. A portion of the membrane pellet was resuspended in 50 mM sodium acetate, pH 5.3, with the aid of sonication (as above) but at a lower power amplitude (19%) and shorter time to create a suspension of fine particles. 3-(N,N-Dimethylmyristylammonio) propanesulfonate (SB3-14) detergent was then added to 0.25% (wt/vol) and mixed by rotation for 24 h at 4°C. The residual membranes were again pelleted as above, and the supernatant was retained as the extracted membrane sample.

**Immunoblot analysis.** Immunoblot analyses were performed as previously described (18, 45) using rabbit polyclonal $\alpha$-Mfa2 antibodies (50).

**Partial deglycosylation.** Portions of the membrane fraction and precipitated soluble fraction were resuspended in 50% acetonitrile–0.1% aqueous trifluoroacetic acid (TFA). The samples were transferred to glass vials, freeze-dried thoroughly, and deglycosylated with trifluoromethanesulfonic acid (TFMS) as previously described (14). TFMS is highly volatile and corrosive and must be used with care in a fume hood. Briefly, samples were placed in an ethanol-dry ice bath for the slow addition of a 50-$\mu$L solution

comprising 90% TFMS and 10% anhydrous toluene. The samples were then transferred to −20°C and the deglycosylation reaction allowed to proceed for 25 min. The reaction mixture was slowly neutralized on an ethanol-dry ice bath with three volumes (150 $\mu$L) of pyridine–methanol–water at a ratio of 3:1:1. Proteins were recovered by the addition of 1 mL distilled water and 200 $\mu$L trichloroacetic acid and centrifugation at 14,000 $g$ for 20 min. The precipitates were washed with ice-cold acetone.

**SDS-PAGE and in-gel digestion.** The extracted membrane sample and the deglycosylated protein samples were separated by SDS-PAGE and fractionated into 8 or 16 gel segments respectively as shown (Fig. 1). Whole cell lysates of the mutant strains were also subjected to SDS-PAGE and divided into 6 gel segments each. The segments were in-gel digested with trypsin as previously described (13) and extracted once with 1% aqueous TFA and once with 50% acetonitrile–0.1% aqueous TFA, both for 15 min in an ultrasonication bath. Extracts were combined, evaporated in a vacuum centrifuge, and dissolved in 2% acetonitrile–0.1% aqueous TFA ready for MS analysis.

**Whole sample trypsin digestion and HILIC.** Another portion of the membrane pellet was run for a short time (~6 min) into an SDS-PAGE gel using all 10 sample wells, stained with Coomassie blue and digested in-gel with trypsin and extracted, all as previously described (51). The extracted peptides were dried in a vacuum centrifuge and then desalted by solid phase extraction using a C18 cartridge. The peptides were eluted in 80% acetonitrile–1% TFA–19% water ready for HILIC enrichment; however, a portion of the peptides were dried and dissolved in 2% acetonitrile–0.1% aqueous TFA ready for MS analysis. ZIC-HILIC stage tips were constructed as previously described in detail (52). The ZIC-HILIC tip was conditioned with 200 $\mu$L of 0.1% TFA followed by 200 $\mu$L of 95% acetonitrile. The sample was then loaded onto the tip and washed with 500 $\mu$L of 80% acetonitrile–1% TFA–19% water. Finally, the peptides were eluted with 150 $\mu$L of 0.1% TFA. Note that the ZIC-HILIC material must remain saturated with solvent at all times throughout the procedure. The eluted sample was dried in a vacuum centrifuge and dissolved in 2% acetonitrile–0.1% aqueous TFA ready for MS analysis.

**Mass spectrometry.** LC-MS/MS experiments were conducted on a Dionex Ultimate 3000 UHPLC interfaced with an Orbitrap Fusion Lumos Tribrid mass spectrometer (Thermo Fisher Scientific) as previously described (53), with the following modifications. For the analysis of the whole sample digests (with or without HILIC enrichment), a long gradient was used going from approximately 2%–20% acetonitrile over 90 min and then up to 32% acetonitrile over 45 min. For analysis of the detergent-extracted sample that was fractionated into 8 gel segments, a shorter gradient of 2%–40% acetonitrile over 60 min was employed. For these analyses of intact glycopeptides, a stepped FAIMS method was employed alternating between −25 V and −45 V. Specific glycan fragment ions (204.0867, 366.1396, and 138.0545 $m/z$) were used to trigger the additional CID, EThcD, and HCD scans (all in the orbitrap) with previously described scanning parameters (53).

For the analysis of acid-cleaved glycopeptides, peptides were eluted over a 30 min gradient from approximately 2%–18% acetonitrile and then up to 32% acetonitrile in a further 10 min. A stepped FAIMS method was employed alternating between −40 V and v60 V, and only HCD spectra were collected using the ion trap. For the collection of ETD and CID spectra, selected samples were analyzed again using the same LC-MS/MS method except that previously identified glycopeptides were preferred by the use of a mass-inclusion list. HCD, ETD, EThcD, and CID scans were all performed in the orbitrap.

To examine the O-glycans present in the mutant strains, the samples were analyzed with a short gradient as for the acid-cleaved samples. For selected gel segments, a method employing only CID fragmentation was used to enable the glycan sequence to be determined. This was followed by the analysis of all the gel segments, employing only HCD fragmentation, and a stepped FAIMS method was employed alternating between −25 V and −45 V.

**Peptide identification.** Proteins and peptides were identified by searching against the *P. gingivalis* ATCC 33277 sequence database (GenBank accession: AP009380) (54). All searches were conducted using trypsin with up to two missed cleavages allowed, peptide mass tolerance was set to 10 ppm, fragment mass tolerance to 0.2 Da, and a fixed modification of carbamidomethyl (C). Initially, the raw MS data were searched with Byonic v3.9.6 (Protein Metrics Inc) using the wildcard setting, initially set to 100–2000 Da and later extended to 2500 Da (55). After identification of most of the glycan additions, the data were searched again using specific modifications as follows: common modifications: oxidation (M), 1394.42 (S, T) and 1436.43 (S, T); rare modifications: 792.25 (S, T) and 1352.42 (S, T). A maximum of 3 common and 1 rare modifications were allowed. For the identification of acid-cleaved glycopeptides, Mascot v2.6.2 (Matrix Science, United Kingdom) was used in error-tolerant mode with oxidation (M) set as a variable modification. Only modification fragments specific to the *P. gingivalis* glycan were added to the list of modifications used for the error-tolerant search. A glycopeptide was considered identified and included in Table S2 if it contained the sequence of the glycosylation motif and was identified by at least one glycopeptide with a Byonic score greater than 200 and −$\text{Log}_{10}$ (P) > 1 or a Mascot score greater than 30. The false discovery rate (FDR) of identifying individual glycopeptides with these thresholds was <0.1% for Byonic. Since decoy mode is not compatible with error tolerant mode, the FDR for the Mascot searches was calculated for the most abundant modifications (Hex-HexA and $\text{Hex}_2$-HexA), which were used as variable modifications. The FDR was insignificant, as no peptides with either modification were identified in the decoy database with a score >30. Where multiple glycopeptides of different sequence or glycan addition were identified for the same site, lower Byonic or Mascot scores were permitted. MS/MS spectral matches relying on a Byonic score between 200 and 300 were manually checked for the presence of signature low mass ions corresponding to HexNAc, strong matches to b or y ion series, and peaks corresponding to glycan cleavages.

**Protein localization.** Proteins were localized as follows. Proteins and lipoproteins were localized to the OM or cell surface according to our recently published study (36). Other localizations were according

to the "gingimaps" study (56) except when some other knowledge indicated a different localization. Lipoprotein predictions were validated by manual inspection of signals, and those that were not known to be OM (36) were not assigned to either membrane since the lipoprotein sorting signal in *P. gingivalis* is not yet known.

**Glycan nomenclature.** The nomenclature used for drawing and abbreviating sugars was taken from the Symbol Nomenclature for Glycans (SNFG) at https://www.ncbi.nlm.nih.gov/glycans/snfg.html (57).

**Data availability.** Proteomics data are available via ProteomeXchange with identifier PXD028120.

## SUPPLEMENTAL MATERIAL

Supplemental material is available online only.

**SUPPLEMENTAL FILE 1**, PDF file, 0.3 MB.

**SUPPLEMENTAL FILE 2**, XLSX file, 0.1 MB.

## ACKNOWLEDGMENTS

We thank Susan Veith for her research assistance, and the staff of the Mass Spectrometry and Proteomics Facility at the Bio21 Institute, University of Melbourne, Australia for the acquisition of Orbitrap LC-MS/MS data and related technical support.

This work was supported by the Australian National Health and Medical Research Council grant ID 1123866 and the Australian Government Department of Industry, Innovation and Science Grant ID 20080108.

Conceptualization, P.D.V., M.S., N.E.S., E.C.R. Formal analysis, P.D.V., M.S. Funding acquisition, P.D.V., E.C.R. Investigation, P.D.V., M.S. Methodology, P.D.V., M.S., N.E.S. Project administration, P.D.V., E.C.R. Resources, P.D.V., M.S., N.E.S., E.C.R. Supervision, P.D.V., E.C.R. Visualization, P.D.V., M.S. Writing—original draft, P.D.V., M.S. Writing—review and editing, P.D.V., M.S., N.E.S., E.C.R.

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
