## [Reviewer comments · Microbiology Spectrum]

Microbiology Spectrum

Characterisation of the O-glycoproteome of *Porphyromonas gingivalis*

Paul Veith, Mikio Shoji, Nichollas Scott, and Eric Reynolds

Corresponding Author(s): Eric Reynolds, University of Melbourne

Review Timeline:

Submission Date:	September 12, 2021
Editorial Decision:	November 12, 2021
Revision Received:	November 25, 2021
Accepted:	December 6, 2021

Editor: Fikri Avci

Reviewer(s): Disclosure of reviewer identity is with reference to reviewer comments included in decision letter(s). The following individuals involved in review of your submission have agreed to reveal their identity: Ashu Sharma (Reviewer #2)

Transaction Report:

DOI: <https://doi.org/10.1128/spectrum.01502-21>

November 12, 2021

Prof. Eric C Reynolds
University of Melbourne
Oral Health CRC, Melbourne Dental School, Bio21 Institute
720 Swanston Street
Melbourne, Victoria 3010
Australia

Re: Spectrum01502-21 (Characterisation of the O-glycoproteome of *Porphyromonas gingivalis*)

Dear Prof. Reynolds:

Thank you for submitting your manuscript to Microbiology Spectrum. We have completed our review of your manuscript, and I am pleased to inform you that, in principle, we expect to accept it for publication. However, acceptance will not be final until you have addressed the reviewer comments.

Link Not Available

Sincerely,

Fikri Avci

Journals Department
Reviewer comments:

Reviewer #1 (Comments for the Author):

Manuscript review - Characterization of the O-glycoproteome of *Porphyromonas gingivalis*

The authors have successfully demonstrated that *P. gingivalis* is capable of O-glycosylating proteins and have characterized the glycoproteome of this bacterium. They also discovered that O-glycosylation is virtually absent on surface-exposed outer membrane proteins, except for a few low-abundance outer membrane proteins with only 1 or 2 glycosylation sites, and they discuss possible explanations for why this may be.

The manuscript is well written and the experiments contained within the paper are also of high caliber.

Comments/items to be addressed

1. Tables 1, 2, 3, and 4 appear to be missing and are not present in the files received for manuscript review? These need to be included in the revision.
2. The authors refer to GalE on lines 80 and 543 as a glycosyltransferase. The enzyme nomenclature is reserved for UDP-galactose-4-epimerases catalyzing the conversion of UDP-Glc to UDP-Gal, so it would be more accurate to write...the GalE putative epimerase (since reference 23 also does not look at enzyme function).
3. Are the authors able to state with any confidence whether or not the glycans are heterogeneously modified with a mixture of P-Gro, P-GroAc, and P-Gro(Ac)₂ vs the presence of unacetylated P-Gro and P-GroAc being present in MS spectra simply a result of fragmentation? It would be good to include a sentence or two in the discussion either way since the presence of this mixture vs only the diacetylated form has implications on the biosynthesis of this glycan.
4. Similar to point 2), are the authors able to infer whether or not the P-Gro* modifications are stoichiometric or not? This would be of interest to readers since many P-Gro modifications on glycans occur in the periplasm and are non-stoichiometric, presumably due to the relative rates of P-Gro transfer vs transfer of the glycan from its acyl carrier.
5. At various points in the manuscript and figure legends, the authors alternate between the terms "gel bands", "gel fractions", "bands", and "fractions" for the same gel segments. The authors should use consistent wording for these throughout the manuscript, and I would suggest they consider using "gel segments" over the above-listed terms, since this minimizes confusion with protein bands or fractions obtained via chromatographies and/or other separation techniques.
6. Line 364, convert HexNac to HexNAc.
7. Lines 594-595. Reword "...and would be tolerated since the abundance is too low to affect..." to "...and might be tolerated due to their low abundance being of little consequence to..." or similar.
8. Figure 1 legend, the authors should spell out TFMS and indicate that the hazard symbol is representative of this acid.
9. The described structure of "Hex-(dHex)-HexA-Hex-dHex-HexNAc-(P-Gro-[Ac]₂)-HexNAc" is ambiguous because it is not clearly indicated to which residues the bracketed residues are linked. It would be more clear to write as "Hex(dHex)-HexA-Hex-dHex-HexNAc(P-Gro-[Ac]₂)-HexNAc" to indicate which residues the bracketed residues are connected to.
10. Also, the linear description is organized in the opposite manner to what is typically used with LPS/CPS, ie. the Hex(dHex) is at the reducing end that would be linked to the protein so this is typically listed last. I would suggest either flipping the order or showing the connection to S/T for clarity.

Reviewer #2 (Comments for the Author):

This is a robust and well described study on the glycosylation aspects of the bacterium *P. gingivalis*. The findings advance our understanding on the O-glycosylation machinery of *P. gingivalis*, and bacteroidetes in general.

Staff Comments:

Preparing Revision Guidelines

Please return the manuscript within 60 days; if you cannot complete the modification within this time period, please contact me. If you do not wish to modify the manuscript and prefer to submit it to another journal, please notify me of your decision immediately so that the manuscript may be formally withdrawn from consideration by Microbiology Spectrum.

Reviewer #1 (Comments for the Author):

Manuscript review - Characterization of the O-glycoproteome of *Porphyromonas gingivalis*

The authors have successfully demonstrated that *P. gingivalis* is capable of O-glycosylating proteins and have characterized the glycoproteome of this bacterium. They also discovered that O-glycosylation is virtually absent on surface-exposed outer membrane proteins, except for a few low-abundance outer membrane proteins with only 1 or 2 glycosylation sites, and they discuss possible explanations for why this may be.

The manuscript is well written and the experiments contained within the paper are also of high caliber.

Comments/items to be addressed

1. Tables 1, 2, 3, and 4 appear to be missing and are not present in the files received for manuscript review? These need to be included in the revision.

The tables are now provided

2. The authors refer to GalE on lines 80 and 543 as a glycosyltransferase. The enzyme nomenclature is reserved for UDP-galactose-4-epimerases catalyzing the conversion of UDP-Glc to UDP-Gal, so it would be more accurate to write...the GalE putative epimerase (since reference 23 also does not look at enzyme function).

Correct, we have replaced “glycosyltransferase” with “putative epimerase” as suggested

3. Are the authors able to state with any confidence whether or not the glycans are heterogeneously modified with a mixture of P-Gro, P-GroAc, and P-Gro(Ac)₂ vs the presence of unacetylated P-Gro and P-GroAc being present in MS spectra simply a result of fragmentation? It would be good to include a sentence or two in the discussion either way since the presence of this mixture vs only the diacetylated form has implications on the biosynthesis of this glycan.

Yes. It is very clear that the P-Gro heterogeneity is genuinely present in the samples as each form elutes at a different HPLC retention time. This means that the various forms are present in the sample prior to their entry into the MS. In order to show this, we present a retention time analysis in a new figure, Fig S2. The data is described by the following new section in the “Results” at lines 461-72: “**Glycan heterogeneity:** Further Mascot searches were conducted to determine whether the phosphoglycerol addition was stoichiometric. Small numbers of glycopeptides with Δ mass values of 995.33 Da and 1198.41 Da corresponding to the inclusion of one or both HexNAc residues without phosphoglycerol were identified. The abundance (peak intensity) of these glycopeptides was typically <1% relative to those with phosphoglycerol suggesting that the modification was nearly stoichiometric. The HPLC retention times of these glycopeptides were always less than their three counterpart glycopeptides substituted with variously acetylated glycerophosphate confirming genuine sample heterogeneity rather than fragmentation within the mass spectrometer (**Fig S2**). Furthermore, acetylation consistently resulted in a slight increase in retention time, again confirming the genuine heterogeneity of acetylation levels (**Fig S2**).”

4. Similar to point 2), are the authors able to infer whether or not the P-Gro* modifications are stoichiometric or not? This would be of interest to readers since many P-Gro modifications on glycans occur in the periplasm and are non-stoichiometric, presumably due to the relative rates of P-Gro transfer vs transfer of the glycan from its acyl carrier.

To address this question, we have performed new Mascot searches to identify a small number of glycopeptides without P-Gro modifications. The peak intensities of these glycopeptides is considerably lower than those with P-Gro such that while the modifications are not stoichiometric they appear to be close to it. See also our response to comment (3) and the new figure (Fig S2) which shows that the glycopeptides without P-Gro are genuinely present in the sample rather than being produced by fragmentation in the MS.

5. At various points in the manuscript and figure legends, the authors alternate between the terms "gel bands", "gel fractions", "bands", and "fractions" for the same gel segments. The authors should use consistent wording for these throughout the manuscript, and I would suggest they consider using "gel segments" over the above-listed terms, since this minimizes confusion with protein bands or fractions obtained via chromatographies and/or other separation techniques.

All occurrences of "bands" or "fractions" pertaining to gel slices have been changed to "gel segments" as requested.

6. Line 364, convert HexNac to HexNAc.

Corrected as specified

7. Lines 594-595. Reword "...and would be tolerated since the abundance is too low to affect..." to "...and might be tolerated due to their low abundance being of little consequence to..." or similar.

This has been reworded as suggested to "...and may be tolerated due to their low abundance being of little consequence to the overall surface chemistry"

8. Figure 1 legend, the authors should spell out TFMS and indicate that the hazard symbol is representative of this acid.

This has been changed as requested to "trifluoromethanesulfonic acid (indicated by the hazard symbol)"

9. The described structure of "Hex-(dHex)-HexA-Hex-dHex-HexNAc-(P-Gro-[Ac]2)-HexNAc" is ambiguous because it is not clearly indicated to which residues the bracketed residues are linked. It would be more clear to write as "Hex(dHex)-HexA-Hex-deHex-HexNAc(P-Gro-[Ac]2)-HexNAc" to indicate which residues the bracketed residues are connected to.

Yes, this has been fixed as requested.

10. Also, the linear description is organized in the opposite manner to what is typically used with LPS/CPS, ie. the Hex(dHex) is at the reducing end that would be linked to the protein so

this is typically listed last. I would suggest either flipping the order or showing the connection to S/T for clarity.

The linear description has been reversed for all as requested to HexNAc-HexNAc(P-Gro-[Ac]₀₋₂)-dHex-Hex-HexA-Hex(dHex) or subsets of this and now match the figure orientations with the reducing end of the glycans at the right side.

Reviewer #2 (Comments for the Author):

This is a robust and well described study on the glycosylation aspects of the bacterium *P. gingivalis*. The findings advance our understanding on the O-glycosylation machinery of *P. gingivalis*, and bacteroidetes in general.

December 6, 2021

Prof. Eric C Reynolds
University of Melbourne
Oral Health CRC, Melbourne Dental School, Bio21 Institute
720 Swanston Street
Melbourne, Victoria 3010
Australia

Re: Spectrum01502-21R1 (Characterisation of the O-glycoproteome of Porphyromonas gingivalis)

Dear Prof. Reynolds:

Your manuscript has been accepted, and I am forwarding it to the ASM Journals Department for publication. You will be notified when your proofs are ready to be viewed.

Sincerely,

Fikri Avci
Editor, Microbiology Spectrum

Journals Department
Supplemental Material: Accept
Supplemental Material: Accept